# Effects of Psychoactive Massage in Outpatients with Depressive Disorders: A Randomized Controlled Mixed-Methods Study

**DOI:** 10.3390/brainsci10100676

**Published:** 2020-09-26

**Authors:** Michaela Maria Arnold, Bruno Müller-Oerlinghausen, Norbert Hemrich, Dominikus Bönsch

**Affiliations:** 1Medizinische Fakultät, Julius-Maximilians-Universität Würzburg, 97070 Würzburg, Germany; dominikus.boensch@bezirkskrankenhaus-lohr.de; 2Berufsfachschule für Massage am Universitätsklinikum Würzburg, 97080 Würzburg, Germany; Hemrich_N@ukw.de; 3Charité, Universitätsmedizin Berlin, 10117 Berlin, Germany; bruno.mueller-oerlinghausen@web.de; 4Bezirkskrankenhaus für Psychiatrie, Psychotherapie und Psychosomatische Medizin, 97816 Lohr am Main, Germany

**Keywords:** massage therapy, psychoactive massage, affect-regulating massage therapy, affective touch, depression, pain, interoception, C-tactile fibers, body psychotherapy

## Abstract

The clinical picture of depressive disorders is characterized by a plethora of somatic symptoms, psychomotor retardation, and, particularly, anhedonia. The number of patients with residual symptoms or treatment resistance is high. Touch is the basic communication among humans and animals. Its application professionally in the form of, e.g., psychoactive massage therapy, has been shown in the past to reduce the somatic and mental symptoms of depression and anxiety. Here, we investigated the effects of a specially developed affect-regulating massage therapy (ARMT) vs. individual treatment with a standardized relaxation procedure, progressive muscle relaxation (PMR), in 57 outpatients with depression. Patients were given one ARMT or PMR session weekly over 4 weeks. Changes in somatic and cognitive symptoms were assessed by standard psychiatric instruments (Hamilton Depression Scale (HAMD) and the Bech–Rafaelsen–Melancholia–Scale (BRMS)) as well as a visual analogue scale. Furthermore, oral statements from all participants were obtained in semi-structured interviews. The findings show clear and statistically significant superiority of ARMT over PMR. The results might be interpreted within various models. The concept of interoception, as well as the principles of body psychotherapy and phenomenological aspects, offers cues for understanding the mechanisms involved. Within a neurobiological context, the significance of C-tactile afferents activated by special touch techniques and humoral changes such as increased oxytocin levels open additional ways of interpreting our findings.

## 1. Introduction

Depressive disorders are among the most common mental diseases in the western world [1]. In addition to the personal suffering of those affected and their environment, the disease represents a significant challenge for society as a whole. Lifetime prevalence has been found to range between 16% and 20%. The World Health Organization (WHO) has declared that it will become the leading cause of global disease burden in the future [2]. The course of the disorder in most patients is episodic and is characterized by a lowered, depressed mood lasting for at least 2 weeks, vital and cognitive retardation, negative thoughts, and the core symptom of anhedonia inflicting nearly all areas of normal life and making life dull and gray. Suicidal ideation and suicidal behavior occur frequently; 10–15% of patients with affective disorders who do not receive efficacious prophylactic long-term treatment (e.g., with lithium salts) will finally die from suicide. In addition, a plethora of somatic symptoms, including pain and physical fatigue affect quality of life, impair working function, increase healthcare utilization, worsen depression outcomes, and increase the risk of recurrence [3]. There is further evidence that depression is associated with substantially disturbed body awareness and desynchronization causing psychomotor retardation [4,5]. These aspects do not receive sufficient consideration if depression is simply called a “mood disorder” or “affective disorder” [6].

It is important to realize that the body (“Leib” in German language and philosophy) of a depressed individual is affected just as much as the mental state. Many facets of colloquial language illustrate the close connection of body and depression, e.g., talking about a person who is depressed or mortified, we may say “he/she is down in the mouth”.

Although the diagnosis and treatment of depression has improved significantly in recent years, there are still deficits in the care and therapy of affected individuals. Optimal treatment success often cannot be achieved, so that in about 30% of cases residual symptoms can be observed. These include primarily, sleep disorders, chronic depressed and/or anxious mood, cognitive deficits, and somatic symptoms [7,8].

The main treatment modalities comprise different types of psychotherapy and/or treatment with antidepressants and other psychotropic agents. However, although prescriptions for antidepressants are rising from year to year in most European countries and in the US [9], their overall efficacy is far from satisfactory. In recent years critical voices based on meta-analyses and serious, independent studies have drawn the attention of a larger audience to the fact that in some patient cohorts the therapeutic efficacy of antidepressants was not found to be higher than that of placebo [10,11,12]. On the other side, increased awareness of patients, doctors, and the general public has been directed particularly to adverse reactions to these compounds, such as a worsened course of the illness, increased cardiovascular mortality, suicidal ideation, or persistent sexual disturbances even after withdrawal [13,14]. Furthermore, it also appears that the beneficial effects of psychotherapy on depression were overestimated in the past [15]. This might be the reason why patients with depression often seek help and relief from their symptoms with “alternative” or complementary therapies [16].

Treatment strategies such as body-oriented methods, including mindfulness-based approaches, have been discussed and explored recently, e.g., body psychotherapy for the management of chronic depression [17]. There is also sufficient evidence for the effectiveness of, e.g., physical training such as aerobic exercise for depression [18,19]. In addition, elements from yoga, tai chi, and qigong are increasingly being investigated and used to treat mental disorders, but evidence of their efficacy is preliminary [20,21].

Survey data suggest frequent utilization of massage therapy or other hands-on treatment among patients with depression [16]. Whether a passive treatment modality such as massage therapy would have a beneficial effect on symptoms of depression presents an intriguing question in view of the great number of patients with treatment-resistant depression [22] and the practical lack of side effects of most types of massage [23]. In medical practice, positive effects of massage therapy, such as relaxation or anxiety relief, and antidepressant effects have been observed. These clinical experiences are supported by a meta-analysis of available studies. On the basis of 10 randomized controlled trials (RCTs) comprising 249 participants, Moyer et al. (2004) found that treatment resulted in a lower post-treatment level of depression than in 73% of the control subjects. The authors considered this finding as persuasive evidence for an antidepressive effect in particular and concluded that the medium effect size equaled that of psychotherapy [24]. In addition, they also found strong evidence for reduced trait anxiety. An earlier meta-analysis by Peters (1999), however, had concluded that existing studies on the effectiveness of massage as a nursing intervention had only limited validity and that more rigorous research would be needed [25]. A meta-analysis of 17 RCTs was performed by Hou et al. in 2010, indicating significant effectiveness in the treatment vs. control groups in spite of only moderate quality of the included studies [26]. Baumgart et al. (2011) presented findings from a systematic survey of 22 carefully selected randomized studies published between 1996 and 2009, including seven studies on patients with depression, anxiety, or exhaustion/fatigue as the main diagnosis [27]. All seven studies, only two of them on hospitalized patients, showed a significant reduction in anxiety symptoms, and depression was significantly reduced in five studies. However, the authors also underlined that the heterogeneity of the diagnoses and the diversity of the controls and the assessment methods still limited valid general conclusions on the efficacy of massage therapy in the treatment of depression.

The website of the American Massage Therapy Association (www.amtamassage.org) clearly documents the still existing paucity of controlled studies in this area and the need for further studies recruiting patients with clinical depression. There is a great variety of massage methods, such as Swedish massage, Esalen massage, Thai massage, etc. [28,29,30]. Our research group has been concentrating for many years on studying the clinical effects of affective touch, also called soft or gentle touch or psychoactive massage, in patients and healthy volunteers. In a previous randomized controlled trial, the application of a specially developed one-hour psychoactive massage (Slow Stroke^®^Massage) showed antidepressive efficacy in hospitalized patients with depression [31]. Sufficient evidence could be provided that within the special setting of this study, gentle touch was the key element to produce the antidepressive/anxiolytic effect on the behavioral and somatic level. In the present study, we investigated the mental and subjective somatic effects of a special form of psychoactive massage, the affect-regulating massage therapy (ARMT) vs. progressive muscle relaxation (PMR) in outpatients with depression. (PMR has become an established relaxation method over the last decades and is already used regularly in the psychiatric and psychosomatic field [32,33]). A direct comparison of the effectiveness of psychoactive massage therapy and PMR in patients with depression has not been done so far. The present study, therefore, tested the following hypotheses:

**Hypothesis 1.** *It is expected that the application of ARMT will prove to be more effective in positively influencing the behavioral and somatic dimensions related to depression than a standardized relaxation method such as PMR. This difference will be reflected in the observer ratings using the assessment instruments described below*.

**Hypothesis 2.** *It is expected that a stronger effect in favor of ARMT will also be seen in patients’ self-assessments. Significant results are expected in the pre-post differences of at least half of the items tested on a specially developed visual analogue scale*.

## 2. Materials and Methods

### 2.1. Study Design

The study was designed as a two-arm, monocentric, randomized controlled intervention study with a fixed number of cases. Every patient in the intervention group received four weekly treatments by means of a standardized massage technique (ARMT). Patients in the control group received four applications of PMR over the same time period. Before starting and immediately after completing the study, patients were assessed by an external rater blinded to the specific treatments the patients had been assigned to. A visual analogue scale was used for self-assessment, which was filled in by the patients before and after each treatment. The study center was the outpatient clinic of the vocational school for massage at the University Hospital of Würzburg (Würzburg, Germany), directed by N. Hemrich.

### 2.2. Instruments for Assessing Mental and Somatic Symptoms

#### 2.2.1. HAMD

The Hamilton Depression Scale (HAMD) was used as the external assessment instrument [34,35]. A total of 17 items concerning the depressive symptoms of the previous seven days are checked. Each item is scored from 0 to 4. The severity of the depression is classified as follows [36]: 0–8 points: no depression or clinically unremarkable or remitted; 9–16 points: mild depression; 17–24 points: moderate depression and ≥25 points: severe depression.

#### 2.2.2. BRMS

The Bech–Rafaelsen Melancholia Scale (BRMS) was used as a further questionnaire for external assessment at multiple time points [37]. It comprises a total of 11 items, which refer to the depressive symptoms of the previous three days. Each item is scored between 0 and 4. The interrater reliability of the German version has been found to be *r* = 0.80 and higher [38]; it has also generally been found to be *r* = 0.80 and higher. The severity of depression is classified according to the sum of all scores as follows: 0–5 points: no depression; 6–14 points: mild depression; 15–25 points: moderate depression and 26–44 points: severe depression.

#### 2.2.3. VAS

A specially designed 100 mm visual analogue scale (VAS) was used for subjective assessment of depressive symptoms consisting of 8 items for self-assessment of the current mood [39]. It was completed by the patients immediately before and after each intervention in order to document changes that occurred during the intervention. Only the symptom of sleep disorder could not be recorded due to the given time frame. In addition, patients could write down free-form personal comments. The following items, indicating the negative poles (zero points) of the scale, were assessed:VAS 1:Stress/tension“I’m very tense”/“I’m completely relaxed”VAS 2:Hopelessness“I feel hopeless”/“I’m full of hope”VAS 3:Internal unrest“I am very restless”/“I am full of inner peace”VAS 4:Pain sensations“I’m feeling pain”/“I’m not feeling any pain”VAS 5:Psychomotor retardation“I feel rigid and immobile”/“I feel light and lively”VAS 6:Tendency to brood“Negative thoughts are circulating in my head”/“I’m thinking positive and optimistic”VAS 7:Loss of drive“I feel limp and listless”/“I feel full of energy and drive”VAS 8:Unpleasant physical sensations“I’m not comfortable in my body”/“I’m comfortable in my body”

#### 2.2.4. Assessment on Clinical Interview

The initial examination included an assessment of the patient’s general history as well as a detailed disease-focused interview. After the end of the treatment series, a final consultation was held. During this semi-structured interview, the patient’s subjective experiences and attitudes toward the study were questioned and documented. All interviews were conducted by the principal investigator (first author) and recorded in writing.

### 2.3. Conducting the Study

#### 2.3.1. Sample Size Calculation

The study by Müller-Oerlinghausen et al. served as the basis for the sample size calculation [31,40]. The mean values and mean differences between pre- and post-assessments and the standard deviation could be derived with regard to VAS. Differences of about 20 scale points with a standard deviation of 25 had been described for several VAS variables. Against this empirical background and assuming a significance level of 5% for two-sided tests, we calculated an optimal sample size of 58.75 cases, which was rounded up to 60. No potential attrition rate was taken into consideration.

#### 2.3.2. Ethical Approval

The study was approved by the Ethics Committee of the Medical Faculty of the Julius Maximilian University of Würzburg. Performance of the study, including informed consent of the patients, followed the Declaration of Helsinki.

#### 2.3.3. Recruitment and Randomization

The patients were recruited in cooperating psychiatric or psychotherapeutic practices in Würzburg. Advertisements were placed in the press and leaflets were distributed at the University Hospital of Würzburg.

Inclusion criteria were as follows:Patients of both sexes between the ages of 18 and 65;The presence of a mild to moderate depressive episode diagnosed by a general practitioner or specialist, including the following ICD-10 diagnoses: F32.0, F32.1, F32.2, F32.8, F32.9, F33.0, F33.1, F33.2, F33.8, F33.9.

Exclusion criteria:Acute comorbid medical condition.Eczematous skin disease;Marked varicose veins or venous thrombosis;Pregnancy;Simultaneous participation in another clinical trial.

The first contact of potential study participants was usually made by phone or email. Suitability for participation in the study was checked on the basis of the inclusion and exclusion criteria. If there was agreement, the patient’s written consent to participate in the study was then obtained. Within two months, 60 patients were recruited. Randomization took place in the order of admission to the study. A randomization list was used, which was based on numbers from the random number generator of the SPSS^®^ statistical program package. After being notified of the randomization result, three participants in the control group terminated their participation in the study prematurely because they did not agree with the assigned group. This resulted in 30 participants in the intervention group and 27 participants in the control group. The initial rating was carried out by an external observer blinded with regard to the randomization result. The threshold for inclusion in the study was set before as 9 points on the HAMD 17 and 6 points on the BRMS. No financial or other compensation was offered to the study participants. The process is shown in Figure 1.

### 2.4. Description of Interventions (ARMT and PMR)

#### 2.4.1. Massage Group (ARMT)

The study was performed in cooperation with the vocational school for massage at the University Hospital of Würzburg. A working group was formed consisting of nine masseurs under training, the principal of the vocational school, and the principal investigator/first author, who is also a certified massage therapist. Through intensive training and communication, a standardized treatment procedure was developed, which was carried out by all therapists equally. Techniques for psychoactive massage described in the literature, e.g., Slow Stroke© Massage [28,31], were taken as the matrix for developing our own massage technique, described in detail below. (To get an idea of these special touch techniques ref. also to some video material under www.bruno-mueller-oerlinghausen.de or www.affective-touch.com).

Any massage session lasted 60 min including rest. It took place in a quiet room. A constant room temperature of at least 25 °C was ensured. The massage oil was preheated to 35 °C. Massage always began with the undressed patient in a supine position. The genital area was covered by a towel. At the beginning, the preheated massage oil was distributed on the ventral body surface to the abdomen, legs, and arms to ensure continuous treatment without interruption. Afterwards, the therapist’s hands rested on the palmar side of the patient’s feet to achieve conscious contact. Subsequently, extensive whole-body strokes were performed. Both hands were used to massage from the legs over the flanks up to the arms and over the pelvis and legs, and back to the starting point. This was followed by treatment of the lower extremities with superficial, partially stretching strokes and soft kneading. The sequence was continued by strokes from the middle of the body cranially and caudally as a diagonal whole-body stroke. After classical abdominal, thoracic, and arm treatment, as well as symmetrical whole-body strokes, the patient changed to the prone position. The treatment then began analogous to the ventral side, starting from the heel bone and continuing over the legs, back, and arms. This was followed by a sequence of diagonal and symmetrical whole-body strokes and soft kneading of the dorsal body. Finally, to mark the end of the treatment, the hands rested on the plantar sides of the feet. The patient was then allowed to rest for another 10–15 min. During this time, he or she was covered with sheets and a woolen blanket. Overall, the treatment was carried out very calmly and evenly. Possible muscular tension was not addressed in order to not interrupt the massage flow by any painful perceptions. No conversations were held during the treatment and no background music was played. Care was taken to ensure that patients were always treated by the same therapist within a treatment series.

#### 2.4.2. Control Group (PMR)

In the control group, progressive muscle relaxation (PMR) according to Jacobson, a widely recognized relaxation method, was used [32,33]. In order to make the general conditions as equivalent as possible to those of the massage group, PMR was not performed in the group, but as an individual 45-min treatment. The instructions were given by the same therapists who were also active in the massage group. At the beginning of PMR, the patient was asked to get into in a comfortable supine position, using a massage table or gymnastics mat as a base. Care was taken to ensure a comfortable position through appropriate room temperature and the use of pillows and blankets. The patient was then individually guided through a PMR whole-body schedule. The standardized instructions were presented in a calm and pleasant tone. Afterwards, the patient could rest for 10–15 min. During the treatment, no conversations were held and no background music was played. Care was taken to ensure that the patients were always treated by the same therapist within one series.

### 2.5. Accompanying Therapy

Any existing therapy with psychotropic drugs and/or psychotherapy could be continued. Changes in this respect were documented at the end of the trial period.

### 2.6. Statistical Evaluation

Absolute and relative frequencies, as well as mean and standard deviation, were used to present personal data such as age, gender, marital status, etc. All data collected manually by the questionnaire were first transferred to an Excel spreadsheet. Data were then analyzed using IBM SPSS Statistics (version 25). Since normal distribution of the parameters could not be assumed, the Mann–Whitney U-test, a nonparametric test, was chosen to calculate statistical significance. The level of statistical significance was set as *p* < 0.05 (two-sided). Pearson’s correlation coefficient r was calculated for estimation of effect size.

## 3. Results

### 3.1. Description of Sample

#### 3.1.1. Sociodemographic Data

The ARMT group consisted of 30 participants. The mean age was 45.2 years; the youngest patient was 24, the oldest, 60 years old; 76.7% were female. About half of the participants were married or in a stable partnership (53.3%). The majority of participants (76.7%) had an intermediate or high school diploma and had qualified employment.

The PMR group consisted of 27 participants. The mean age was 45.0 years; the youngest patient was 19, the oldest, 64 years old; 81.5% of were female. About half of the participants were married or living in a stable partnership (55.6%). Almost all participants (96.3%) had an intermediate or high school diploma and had qualified employment; in this respect they were somewhat different from the subjects of the ARMT group. The sociodemographic data is shown in Table 1.

#### 3.1.2. Depression-Related Data

In the ARMT group, the severity of depression was rated as moderately severe, corresponding to 18.2 points on average on the HAMD at the beginning of the study. Assessment by means of the BRMS (mean value 16.1 points) also confirmed moderately severe depression. Psychopharmacological treatment as mono- or combination therapy was reported by 56.7% of the participants. There was no change in dosage for 83.3% of the patients over the duration of the study. About half (53.33%) of the participants were under psychotherapeutic treatment during participation in the study.

In the PMR group, the severity of depression rated by means of the HAMD averaged 19.2 points at the beginning of the study, also indicating a moderately severe depressive episode. The BRMS scores suggested an identical classification, with an average of 17.4 points. Psychopharmacological treatment as mono or combination therapy accounted for 55.5% of the participants. There was no change in dosage for 85.2% of the participants during participation in the study. Almost half (48.2%) of the participants were under psychotherapeutic treatment during the study.

### 3.2. Effects of ARMT and PMR Assessed by HAMD and BRMS

The averaged differences between the HAMD assessment at time *T*A, before the first treatment, and time *T*B, after the fourth treatment, were calculated. The reduction in symptom burden over 4 weeks was significantly more pronounced in the ARMT group than in the control group (*p* = 0.034, *r* = 0.28). The results are shown in Figure 2.

Focusing on individual items of the HAMD, changes in the following items proved to be highly statistically significant with a moderate effect size:HAMD 1: Depressive mood (*p* = 0.004, *r* = 0.39)HAMD 13: Somatic symptoms (*p* = 0.021, *r* = 0.30)

Although statistical significance was not reached, we consider the numerical change of item five (sleep disorders) to be particularly noteworthy (*p* = 0.059, *r* = 0.25). The course of the BRMS scores is depicted in Figure 3. The reduction in symptom burden during 4 weeks was significantly more pronounced in the massage group than in the control group (*p* = 0.04, *r* = 0.27). Focusing on individual items, the difference of the following single BRMS dimensions over time proved to be statistically significant:BRMS 8: Emotional retardation (*p* = 0.037, *r* = 0.28)BRMS 9: Sleep disorders (*p* = 0.038, *r* = 0.28)

Although statistical significance was not achieved, special attention might be given to the change in item one, motor activity (*p* = 0.063, *r* = 0.25).

In summary, the findings so far are in accordance with Hypothesis 1 (see above).

### 3.3. Results of Participant’s Self-Assessment (VAS)

For each VAS item (one to eight), the pre-post differences related to each treatment session were averaged over time points *T*1–*T*4. Figure 4 shows that the treatment effects were markedly and significantly more pronounced in the ARMT group as compared to the PMR group. Statistical significance of this difference existed for six of the eight items. (Descriptions of single items can be found in Section 2.2.3).

Please note that a value of 100 on, e.g., VAS 1 (“tension”) would signify that the proband felt completely relaxed. (A score of 100 is always the positive pole and zero the negative pole of each VAS item.) In detail, the following individual VAS items show (highly) significant changes in favor of the massage treatment, i.e., greater pre-post differences of particularly those variables closely associated with depression:VAS 1: Stress/tension (*p* = 0.035, *r* = 0.28)VAS 2: Hopelessness (*p* = 0.032, *r* = 0.28)VAS 3: Internal unrest (*p* = 0.009, *r* = 0.35)VAS 4: Pain sensations (*p* = 0.003, *r* = 0.39)VAS 5: Psychomotor retardation (*p* = 0.012, *r* = 0.33)VAS 8: Unpleasant physical sensations (*p* = 0.011, *r* = 0.34)

A moderate effect size was observed for most of the changes. The pre-post differences in VAS 6 (tendency to brood/negative thoughts, *p* = 0.075, *r* = 0.24) and VAS 7 (lack of drive, *p* = 0.070, *r* = 0.24) were not different between the two treatment groups. To illustrate the therapeutic process, Figure 5, as an example, presents the time course of VAS 5, psychomotor retardation (pre- and post-treatment scores) over the four treatment sessions. Obviously, the treatment effects are more marked in the ARMT group.

Consequently, Hypothesis 2 could be confirmed, i.e., it is expected that a stronger effect will also be seen in the patients’ self-assessment. Significant results are assumed in at least half of the VAS items.

### 3.4. Statements of Study Participants in Clinical Interviews

The statistical results are underlined by the personal statements of the study participants. Particularly frequent positive comments were made by patients assigned to the massage group:Better body awareness and deep relaxation;Interruption of brooding and negative thoughts;Increased motivation for everyday life activities.

As points of criticism, some patients mentioned a feeling of coldness and a sense of shame about taking off their clothes before the massage.

Additionally, patients in the control group described positive effects of PMR:Useful in everyday life and as a sleeping aid;Easy relaxation and better body awareness.

The lack of background music was occasionally mentioned as a point of criticism. Patients in both groups indicated that they had particularly benefited from the morning sessions. The early treatment appointment significantly reduced the burden of a matutinal depressive mood and had a positive effect for the rest of the day. Some participants also experienced a continuous increase in the perceived positive effects during the course of the study. As a point of criticism, study participants from both groups mentioned particular unfavorable external conditions, such as noise from the adjacent construction site.

## 4. Discussion

As outlined above, somatic symptoms such as psychomotor retardation, sleep disorders, or general fatigue are prominent features of the clinical picture of depression. Thomas Fuchs perceives depression as a physical illness existing not only against the background of functional disorders of the entire organism, such as dysregulation of the hormone system and biorhythmics, metabolic and immunological changes. Rather, as a phenomenologist, he describes a disorder of the basic physical constitution that presents itself as a “corporisation of the body” (ref. [41] the contrasting significance of “corpse” and “body” in English). Like an objectification, the patient experiences himself distanced from his body-self and no longer “at home in his body”, as the sociologist Hartmut Rosa illustrates it in general terms. The depressed person does not feel “comfortable in his skin”, where the skin in Rosa’s terminology constitutes an organ of resonance [42]. This disturbed bodily feeling manifests itself in different regions, such as tightness in the chest, heaviness of the limbs, or chronic fatigue. The body loses drive and spontaneity. Everything feels heavy. This is also reflected in the subject’s exchange with the environment: breathing is flattened, facial expressions are reduced, libido is often tuned down. The “feeling of numbness” overshadows every perception, so that even crying is no longer possible. It is not mourning that is felt, but the feeling of emptiness and lifelessness that is expressed primarily in the body. “Being a body is replaced by having a body” [43]. Would it not be possible that the cognitive and emotional characteristics of depression are secondary reactions to the perception of the primary bodily changes [44,45]?

Hartmut Rosa tries to separate the depressive experience from grief: “Grief is an element of a relationship to the world that is, on the whole, quite resonant. […] Depression, on the other hand, is characterized by the fact that there are no more tears: the relationship to the world can no longer be liquefied, it is petrified” [42]. The patient feels unable to counteract this weighing feeling of heaviness, and this fundamental phenomenon might induce speculation on a potential bridge to the concept of “learned helplessness” [46].

Against this background, it seems a rational approach to use body-oriented therapies in the treatment of depression. Our findings indicate antidepressive, anxiolytic, and analgesic effects, which were significantly more pronounced among the participants in the massage therapy group as compared to those in the PRM group. How can these remarkable therapeutic effects of a professional, empathetic, affective touch technique be explained? We shall discuss some options on various explanatory levels.

One of the most obvious effects of massage therapy, according to the patients’ self-assessment, is the pronounced psychomotor relaxation, which is also reflected in the reduction in inner restlessness. However, it was not only general tension that was released by the treatment. Rather, feelings of hopelessness and inner restlessness were also significantly reduced. The therapeutic touch technique also had a positive effect on the existing obsessive brooding tendency. In the final open talks with the patients, we often heard statements such as that the massage finally allowed the individual to “turn off my thoughts” or “to break loose”. One patient made a written comment after the third massage session: “Never before did I experience such a deep relaxation. None of the usual relaxing (mind–body) exercises had this strong effect”.

However, even if this feeling of relaxation is a significant part of the overall effect, there must be other factors involved, otherwise the superiority of massage over a standardized relaxation method would hardly make sense. In modern theories of depression, the concept of interoception has been given increasing attention [47,48]. Interoception, i.e., the perception of the processes of the body’s interior, distinguishes between proprioception (perception of body position and movement in space) and visceroception (perception of the inner organs’ activity). In contrast to exteroception, the signals making up interoception are sent from the entire inner milieu of the individual to the brain. Interoception can likewise be understood as a skill that can be trained through regular practice (e.g., in mindfulness methods), thus contributing to a more conscious body perception and better emotional self-regulation.

In order to explain the effects of the body-oriented interventions examined in this study, we may refer to this concept. The external conditions per se that were more or less concordant in both groups provided a framework for increased body perception. A calm atmosphere without distractions was created to enable the patients to experience consciously their bodily condition, in the sense of an “inner view”. The special feature of affective touch is the calm, mindful approach enabling patients to consciously experience their body. This procedure can thus be described as an intensive training of interoception. The patients’ oral statements underline this assumption.

Recently a hypothesis was put forward that the antidepressive effect of affective touch can be explained by a normalization of disturbed interoception [49]. Another special factor that could explain the significant superiority of affect-regulating massage based on its basic element, affective touch, is the factor of touch itself. A gentle, empathetic touch is generally experienced as pleasant. It can soothe feelings of social exclusion and facilitate interpersonal binding [50,51]. The neurophysiological correlates of this type of touch have been intensively investigated during the last two decades. The specific feeling of well-being that humans and hairy mammals experience with this type of touch is based neurophysiologically on the activation of so-called C-actile afferents. In particular, Swedish researchers were able to show that a neuronal network of slowly conducting, non-myelinated C-fibers reacts to special receptors of the hairy skin (located, e.g., on the back of the hand, but not on the palm). For these receptors, gentle, slow, and rhythmic touch at a speed of about 3 cm/sec is the appropriate stimulus, projected directly and predominantly into the insular area. The stimulation of such C tactile fibers seems to have as its only “purpose” creating a feeling of well-being [52]. This might also have evolutionary biological significance, e.g., by promoting the feeling of kinship within a group. Other afferent projections via A-beta or A-alpha fibers as well as signals from mechanoreceptors in the hairy skin most likely will also contribute to the overall interoceptive signaling taking place in various limbic structures.

Switching to the level of experience and behavior, it should be kept in mind that touch is the basal medium of communication among humans and animals [53,54]. Against the background of anhedonia, described as the most common feature of depressive experience and behavior, our findings are in some way “paradoxical”. How can patients with depression feel and express bodily and verbally affective touch as a positive experience? Obviously empathic, professional touch can slip into this basic communication disorder, i.e., can enter the disturbed emotional world of the patient. It fits into this concept that participants in the massage group often expressed feelings of having been “accepted” in the final talks with the first author. The following English translation of a patient’s original final statement might help us in understanding some essentials of the therapeutic process:

“I cried a lot the first time. For the first time I was able to feel my legs and my feet. Tears of joy during the treatment. From the second session onwards, even better with a warming pad. Meeting of the hands on my belly—very touching. I never experienced such a loving touch before. With further treatments I was loosening up completely. Already during the second session I perceived more than the first time. Even my hands were treated! Because of the depression I couldn’t allow touching me otherwise.... My shell has become softer, my heart is opened. The massage therapist was totally super gentle, helped me to become a ”whole person” again”.

Even though the patient quoted above was in all probability not familiar with the work of Wilhelm Reich [55], in her description she nevertheless takes up an essential aspect that plays an important role in modern body psychotherapy [56]. In Reichian terminology, one can speak of an armoring of the emotions [55], which can be influenced, if not eliminated, by massage therapy. According to our observations, patients often are taken by surprise that after a massage session their body feels “less heavy”. A trusting therapist–patient relationship in the field of physiotherapy seems to be just as important as is already recognized in psychotherapy. The therapeutic relationship thrives on appropriate empathy and the right balance between affectionate empathy and professional demarcation. In contrast to psychotherapy, however, in the physiotherapeutic context it is specifically the body of the patient that is perceived and accepted and thus enters into resonance with the therapist. This leads to new, possibly corrective (body) experiences, which can be of decisive importance for the therapeutic process. To quote the clinical psychologist C.A. Moyer from his meta-analysis of studies on the effects of massage therapy in depression: “The finding that massage therapy has an effect on trait anxiety and depression that is similar in magnitude to what would be expected to result from psychotherapy suggests the possibility that these different treatments may be more similar than previously considered” [24].

However, coming back to the mechanism involved, a decisive difference exists in healing touch: according to Changaris, we offer the patient direct affect-regulating “bottom-up” therapy in contrast or addition to the “top-down” technique of cognitive psychotherapy [57]. Finally, humoral effects such as increased oxytocin levels and reduced cortisol, e.g., in saliva might add to the interpretation of the antidepressive effects of psychoactive massage within a neurobiological context [58,59]. Oxytocin has been attributed with significant effects on social interaction, as well as feelings of trust and connectedness [60]. Thus, it does not seem unlikely that depressive patients suffering from social withdrawal and isolation could profit from increased oxytocin release, possibly facilitating contact and communication [61]. Additionally, the analgesic effects of massage therapy might be related to increased oxytocin release [62]. Tiffany Field [58], against the background of various experimental studies, has often emphasized the importance of markedly reduced cortisol levels in urine or saliva of various diagnostic groups of depressed or stressed patients having been given massage therapy. The strong relaxing effect observed also in our patients might well be related to this hormonal change. Tiffany Field, however, has often argued that the lowered cortisol levels are related to increased vagal activity [63].

### 4.1. Strengths of the Study

#### 4.1.1. Control Group

It was already pointed out that the choice of an adequate control group, besides the impossibility of blinding, is one of the biggest methodical problems in developing a meaningful design in massage studies. Often the control conditions are chosen in such a way that only a standard therapy, waiting list, or general relaxation (quiet sitting/lying, relaxing music or movies) is used. In our study, a more adequate procedure was chosen instead, which led to the most equal conditions possible for all study participants, with the essential difference that therapeutic touch occurred only in the massage group. This allowed us to strengthen the evidence that it is not a bunch of mostly unknown unspecific effects such as personal devotion, but the affective touch as such that is responsible for the greater effectiveness of massage therapy. “Adequate control conditions” signifies that participants in both the massage and control groups always received individual treatments. PMR is typically performed as group therapy. It can be assumed that performing PMR as an individual treatment increased its effectiveness. In other words, we selected a rather conservative approach, which resulted in moderate effect sizes. Use of a “placebo” control would most likely produce much greater effect sizes. In addition, an identical group of therapists were involved in both groups in order to avoid distorting personal influence. Furthermore, the same premises were used for both the massage and control groups. All treatments took place in comparable periods within four months and in a constant environment.

#### 4.1.2. Study Participants

Relatively few studies on the efficacy of massage therapy have been conducted so far in patients with mental illness. The present study provides further evidence of the positive effects of massage therapy in depression. In selecting the study participants, we focused on patients with mild to moderate depression whose clinical picture and disease severity reflected a broad section of the general population. In this way we were able to create largely realistic conditions as they are encountered daily in practices of both GPs and psychiatrists, but also physiotherapists.

#### 4.1.3. Assessments

Standardized and widely used psychiatric scales for assessing and quantifying depressive symptoms were employed in validated German translation.

#### 4.1.4. Conducting the Study

In both groups, care was taken to ensure the consistency of therapists in individual treatment series. In addition to reducing variance, this also served to maximize the effects, as other studies have shown the importance of therapist consistency [64]. Furthermore, all therapists were equally prepared for the study owing to intensive training. This ensured a standardized and comparable execution of individual therapies.

### 4.2. Limitation

One important criterion for the quality of studies contributing to the bulk of evidence-based medicine, besides randomization and active control, is the blinding of study participants and directly involved investigators. For obvious methodological reasons, this requirement could not be met in the present study, as in many comparable studies. However, according to a recent meta-analysis, the absolute postulate of blinding when carrying out a sound study might be somewhat questioned in the future [65]. Furthermore, the self-assessment questionnaires (before and after each treatment) were handed out to the patients by the treating therapists themselves. In retrospect, the critical question came up as to whether this circumstance may have led to distortion. One may speculate whether it would have been important for some patients to leave a positive impression or not to “disappoint” the therapist by filling out a questionnaire in a neutral or negative way. However, since the same questionnaire handling was used in both groups, this possible bias was not further discussed, which might have caused a greater overall effect, but probably did not distort the difference between the two groups

### 4.3. Outlook

Due to the limited effectiveness of the currently available and widely used treatments in depression, it seems reasonable to expand the therapeutic spectrum for inpatients and outpatients with body-oriented procedures. In this context, affect-regulating or comparable psychoactive massage therapies represent a noteworthy opportunity to open up new access routes for acute treatment. They can be used as low-threshold offerings in the outpatient setting in order to achieve a rapid antidepressive effect. Their use in an inpatient setting is also conceivable according to the findings of a previous controlled study [31]. Very good adherence can be expected, as the present study was also able to prove. Body-oriented therapeutic approaches should be given higher value within the spectrum of antidepressive treatments. They also deserve a special place beside much propagated mind–body techniques [66,67]. A severely depressed patient will often be unable to participate in special psychological training sessions aimed at stress reduction.

## 5. Conclusions

This randomized controlled intervention study examined the psychophysical effects of body-oriented treatment methods on patients suffering from mild to moderate depression in an outpatient setting. We tested the hypothesis that a one-hour massage based on a special gentle touch technique (affect regulating massage therapy = ARMT) is superior in its positive effects to a relaxation method that has long been established in the clinical field, progressive muscle relaxation according to Jacobson. Our results confirm this assumption. In both the observer ratings and self-assessments of patients using a visual analogue scale, statistically significant superiority of massage therapy was shown. When focusing on individual dimensions of the HAMD, the superior effects were particularly evident in the items “depressive mood” and “general somatic symptoms”. Assessment using the BRMS showed statistically significant superiority of massage therapy particularly in the items “emotional retardation” and “sleep disorders”. As for the sleep disorders so often encountered in patients with depression, it should be noted that massage had a positive effect, especially on difficulty remaining asleep, while difficulty getting to sleep responded better to PMR. This observation also coincides with the free-form statements of some study participants who used the PMR technique outside of the study, independently as an aid to fall asleep. Especially in the self-assessment (VAS) of patients, massage therapy proved to be superior. This became obvious for the vast majority of items questioned. When inspecting the pre-post differences of individual VAS items, the stronger impact of massage therapy on the dimensions stress/tension, internal unrest, unpleasant physical sensation, psychomotor retardation, and hopelessness is particularly impressive. Changes were also marked for pain sensations. Overall, we were able to document the doubtless superiority of ARMT for core symptoms of depressive experience and behavior.

## Figures and Tables

**Figure 1 brainsci-10-00676-f001:**
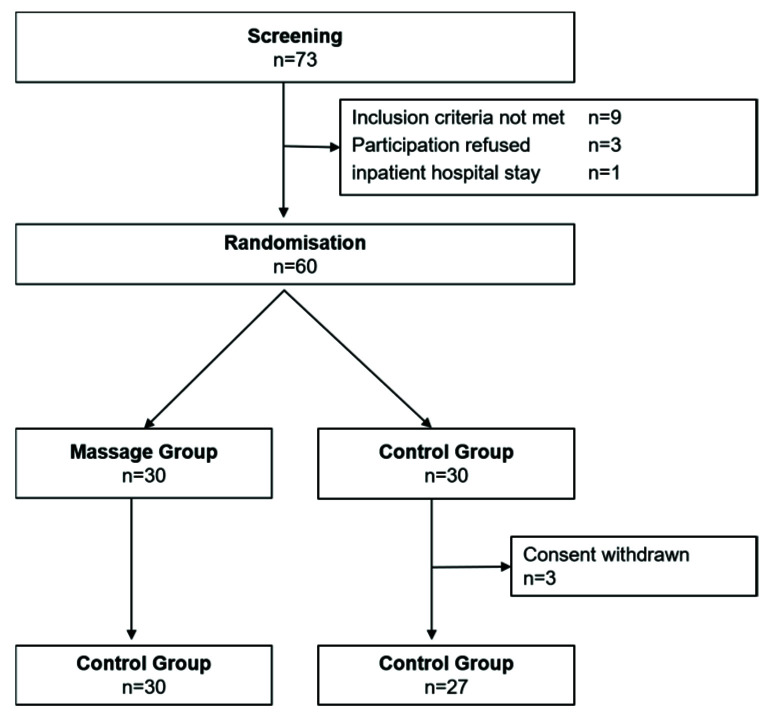
Recruitment and randomization.

**Figure 2 brainsci-10-00676-f002:**
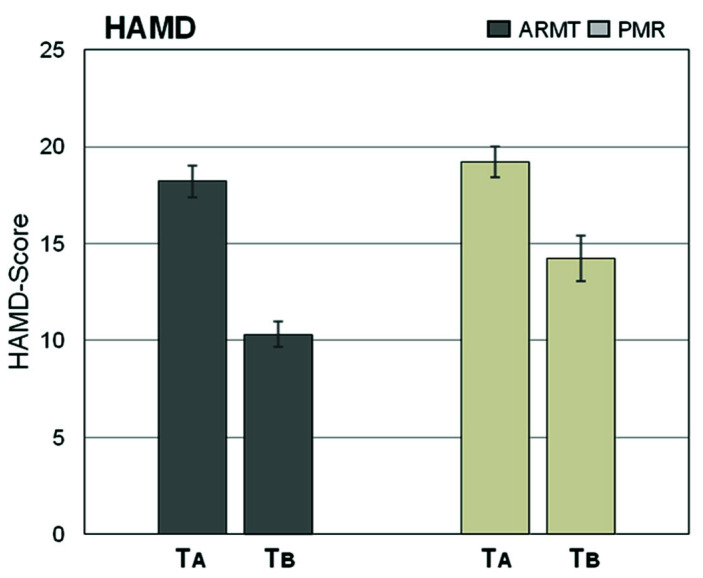
Summed up Hamilton Depression Scale (HAMD) scores in ARMT and PMR groups before (*T*A) and after (*T*B) completing the full series of treatments (mean ± SEM).

**Figure 3 brainsci-10-00676-f003:**
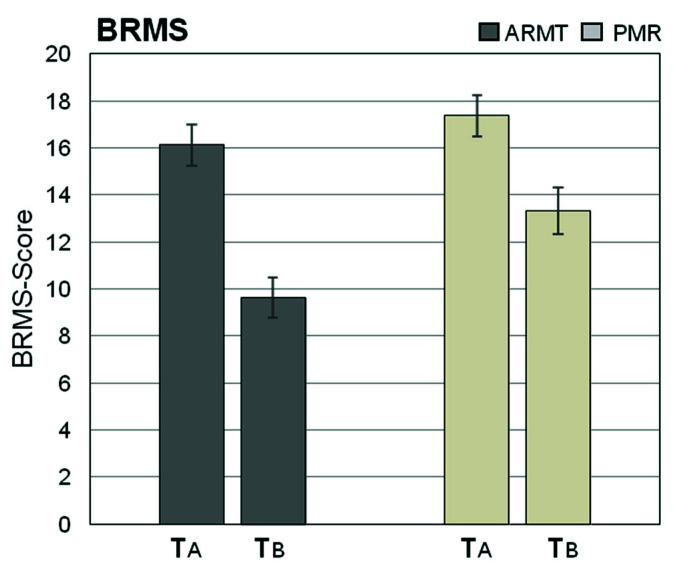
Summed up Bech–Rafaelsen Melancholia Scale (BRMS) scores in ARMT and PMR groups before (*T*A) and after (*T*B) completing the full series of treatments (mean ± SEM).

**Figure 4 brainsci-10-00676-f004:**
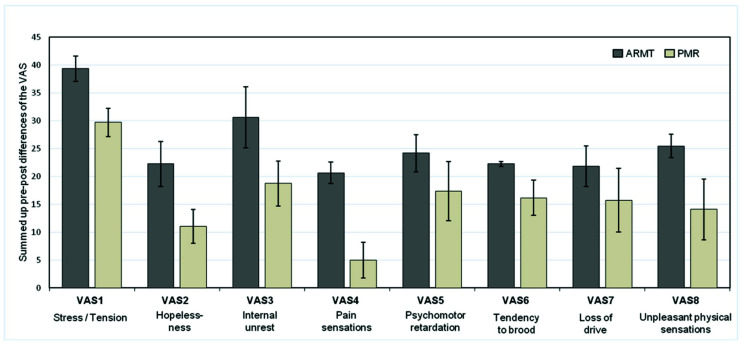
Averaged differences (± SD) of individual pre-/post-treatment visual analogue scale (VAS) values over total study period (*T*1–*T*4).

**Figure 5 brainsci-10-00676-f005:**
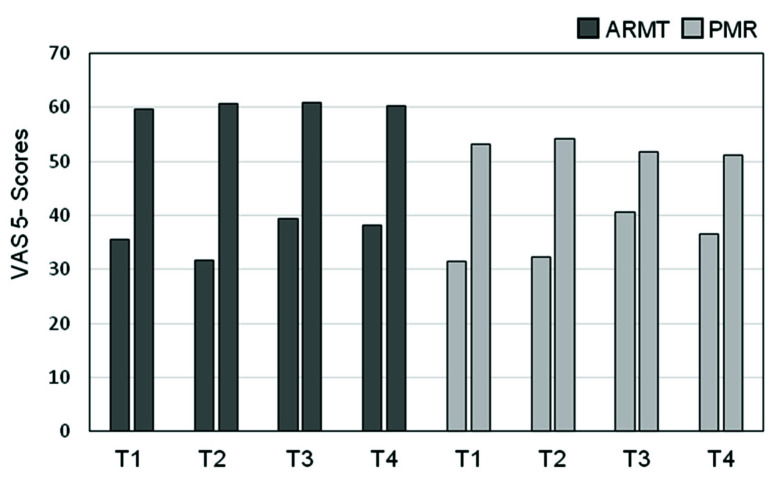
Time course of VAS 5 (psychomotor retardation) scores over time points *T*1–*T*4 in ARMT vs. PMR groups. Arithmetic means before and after treatment (right and left columns, respectively). Note: A score of 100 signifies that the proband agrees fully with the statement “My body feels light and mobile”.

**Table 1 brainsci-10-00676-t001:** Sociodemographic variables of the two patient groups.

	ARMT	*n* = 30	PMR	*n* = 27
	*M*	*SD*	*M*	*SD*
**Age** (years)	45.2	9.43	44.9	12.29
**Sex**	***n***	**%**	***n***	**%**
female (n)	22	73.33	22	81.48
male (n)	8	26.67	5	18.52
**Civil status**				
single	9	30.00	9	33.33
married/partnership	16	53.33	15	55.55
divorced/widowed	5	16.67	3	11.11
**Children**				
childless	11	36.67	15	55.55
one or two children	13	43.33	6	22.22
three or more children	6	20.00	6	22.22
**Education**				
primary school	7	23.33	1	3.7
secondary school	13	43.33	25	92.6
high school	10	33.33	1	3.7
**Employment**				
vocational training	23	76.67	22	81.48
academic career	7	23.33	5	18.52
unemployed	6	20.00	7	25.92

ARMT = affect regulating massage therapy; PMR = progressive muscle relaxation; M = Mean; SD = Standard Deviation.

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
