# Peer review of "Effects of Psychoactive Massage in Outpatients with Depressive Disorders: A Randomized Controlled Mixed-Methods Study"

_brainsci, 2020, doi:10.3390/brainsci10100676_

Round 1

Reviewer 1 Report

The present study was aimed to evaluate the benefits of psychoactive massaging in reducing the symptoms associated with moderate depression. There has been a long effort to understand the basic neurobiology of touch as an intervention to treat the mental health issues especially, the stress associated mood disorders. It has been suggested that various forms of massage have diverse impacts on brain circuitry or brain regions. Even neuroimaging studies have presented favorable evidence of massage on the brain functionality. So far it has been an area of increasing interest but unfortunately largely remain unexplored. However, in the present study the authors have made good initiative to investigate the effects of a specially developed affect regulating massage therapy (ARMT) as an antidepressive/anxiolytic intervention in clinically depressed patients. The encouraging results from this study stated a clear and statistically significant superiority of ARMT vs. PMR or Progressive Muscle Relaxation, a standardized non pharmacological method to relax the muscle tone mostly found effective to control stress and anxiety. However, the reviewer is concerned about five major points that need to be addressed by the authors.

  1. In the abstract, the authors mentioned the changes in oxytocin as possible underlying mechanism of this ARMT therapy. However, later in the manuscript, they have largely ignored the role of vasopressin and adrenocorticotropic hormones.
  2. In the results, the reviewer was expecting to see data from time-point measurements of cortisol level before and after massage interventions.
  3. The concept of interception has been introduced in the discussion. However, the authors did not discuss about the neural pathways associated interception. There are several integral connections from autonomic afferent to central nervous systems which have been largely implicated in neural processing of interceptive signals. The authors are advised to carefully present this missing part.
  4. It has been suggested in the past that increased parasympathetic tone due to increased activity in the forebrain-amygdala system might be associated with positive affect due to massage therapy. The authors are suggested to comment on this.
  5. The c-tactile afferents are associated with positive affect besides A-beta and A-alpha afferent projections. The c-tactile fibers have projections to the limbic and integrate tactile and sensory information from mechanoreceptors of hairy skin surfaces. How the authors are concerned about this in relation to their findings?

Minor points:

  1. The abbreviation of CT should be expanded in the abstract.
  2. The discussion needs to be cut down, at this point it more looks like a story telling type.

Author Response

Response to Reviewer 1

We thank the reviewer for the comments and detailed proposals to amend this article. We accept many of the critical  statements. In the following we are going to describe the changes we have made and will also give our arguments where we were not able to follow the reviewer’s recommendations.

We would like to underline that the present version of the manuscript has been edited by a professional Editing Service of MDPI using British English. We can add the certificate of this service if needed.

  1. This paper – in contrast to many other publications in this field - has laid some emphasis on mechanisms of massage which can be described as phenomenological or psychological. (We regret if our discussion partly has triggered the impression of the authors as “story tellers”…) In order not to overload the paper we were somewhat restrictive in the discussion, knowing that there exist many and highly interesting biological aspects  which are given attention in many other papers. We have added in line 511 a sentence giving more emphasis to the humoral changes. We are, however, also aware that reviewer 1 and 2 both criticize the length of the discussion. In our view this is a paper which in contrast to many other articles deals with basic emotional and social aspects in order to understand better the way depressed patients perceive this special kind of treatment.
  2. Sorry, as we just explained this study was not focused on biochemical changes . No hormone assessments have been performed.
  3. We feel that in our description and discussion of the interoception concept we did already put sufficient emphasis on the neurobiological mechanisms involved, ( line 428 – 459) In the quoted papers the interested reader will find much more detailed information. Since both reviewers uttered some criticisms on the length of the paper we were anxious not to overload it even more.
  4. The question of vagal activity is an interesting one. Tiffany Field has often expressed the view that the effects of massage on e.g. the immune system are related to increased vagal activity which according to her own studies is only triggered by massage using moderate pressure. By the papers of Forbes vagal tonus has created much discussion also in view of changes of heart variability etc. (ref. however, Grossman’s detailed critic of this very special view…). We do not think that we could into detail here. We did allude to this aspect on line 510 referring also to another paper by Field et.al. (63)
  5. C-tactile afferents and other fibers: We added a sentence in line 460-463.

Minor points:

1.) done.

2.) The reviewers are critical about the overall length of the discussion, which by the   way would be even longer in case we would start it with a summary of the      results. We have therefore inserted some more breaks between the various         aspects of mechanisms involved. We have also seriously checked whether   complete parts of the discussion could be removed. However we had to            conclude that this would seriously destroy the inner structure of our             argumentation. But having it split up in now  11 sections we feel that the text       will be better digestible.

Thanks again for  the careful reviewing of our paper,- in which we have put special effort  in order to make the antidepressive effects of therapeutic touch better understood, in our common effort to improve the treatment of our depressed patients.

Reviewer 2 Report

The study addresses an important topic—the application of psycho-regulatory / psychoactive massage—as an adjunct treatment for moderate depression. This is an important area of research because body therapies add an important component to treatment and have thus far been ignored. Whilst the findings of the study are interesting, the manuscript has many inconsistences and problems (for example paragraphs that are so long that they are indigestible for the reader) and these need to be fixed. I have tried to note the various problems to help the authors fix them. I have to admit however that by the discussion I ran out of steam. The discussion also needs work for the manuscript to be brought up to publication standard.

The actual content of the study is important however and I think it is well worth fixing the manuscript problems to get the study out.

Abstract:  RE: The clinical picture of depressive disorders is characterized by a plethora of somatic  symptoms, psychomotor retardation and particularly anhedonia.

There is a problem throughout the manuscript that the authors need to sort out.

They need to sort out the use of somatic symptoms: do they mean the somatic symptoms that accompany depression. If so then this means to be made more clear.

They need to sort out when they use the term psychosomatic symptoms or the more recent term functional somatic symptoms. The terms are not used consistently throughout the manuscript.

Also in the first line of the abstract, I don’t think anhedonia should be included as a somatic symptom. Anhedonia is a loss of interest and pleasure in activities.

Abstract

 the significance of the CT afferences: the term CT cannot be used in the abstract because it has not been defined.

Line 39

In English and current diagnostic systems depression is, I think, referred to as a disorder or an illness (rather than disease). The authors should check this in ICD which is what they use. I notice that a few lines later illness is used.

The term disease burden is fine.

Line 47, not sure about journal style, but most journals do not allow you to start a sentence with a number. It makes the text very hard to read.

Line 40 : put (e.g. with lithium salts) in a bracket. The sentence is otherwise difficult to read for lack of punctuation.

Line 49 RE “a plethora of somatic and psychosomatic symptoms”

Maybe the authors could replace this with “a plethora of functional somatic symptoms” which seems to be the current term that is used as an umbrella term for all of these symptoms. See

Roenneberg, C., et al. (2019). "Functional Somatic Symptoms." Dtsch Arztebl Int 116(33-34): 553-560.

                BACKGROUND: Approximately 10% of the general population and around one third of adult patients in clinical populations suffer from functional somatic symptoms. These take many forms, are often chronic, impair everyday functioning as well as quality of life, and are cost intensive. METHODS: The guideline group (32 medical and psychological professional societies, two patients' associations) carried out a systematic survey of the literature and ana- lyzed 3795 original articles and 3345 reviews. The aim was to formulate empirically based recommendations that were practical and user friendly. RESULTS: Because of the variation in course and symptom severity, three stages of treatment are distinguished. In early contacts, the focus is on basic investigations, reassurance, and advice. For persistent burdensome symptoms, an extended, simultaneous and equitable diagnostic work-up of physical and psychosocial factors is recommended, together with a focus on information and self-help. In the pres- ence of severe and disabling symptoms, multimodal treatment includes further elements such as (body) psychotherapeutic and social medicine measures. Whatever the medical specialty, level of care, or clinical picture, an empathetic professional attitude, reflective communication, information, a cautious, restrained approach to diagnosis, good interdisciplinary cooperation, and above all active interventions for self-efficacy are usually more effective than passive, organ- focused treatments. CONCLUSION: The cornerstones of diagnosis and treatment are biopsychosocial ex- planatory models, communication, self-efficacy, and interdisciplinary mangagement. This enables safe and efficient patient care from the initial presentation onwards, even in cases where the symptoms cannot yet be traced back to specific causes.

Line 53 RE: “to the subject of the present study”

It is tricky to say this at the current point because the subject has not been articulated yet so the reader does not know what the authors mean. Maybe the authors need to be more explicit.

Line 62 RE somatic symptoms, the term functional somatic symptoms would be more clear. Most of the somatic symptoms in patients with depression will not have an identified organic cause. See Roemmeberg above.

Line 64 Re: “psychotherapeutic methods”

I was unsure what this meant. Do the authors mean psychotherapy, different types of psychotherapy?

Line 69 re “many patient”

I wonder about saying some patient…because different trials show different results.

Line 73 “increased cardiovascular mortality, bleeding..

I think the authors need to be very careful here. Depression in itself is a risk factor for cardiovascular disease. The underlying mechanisms are likely very complex. Not surprising that the risk is still increased even after withdrawal of antidepressants. See Rajan et al 2020 for example. Making simplistic associations is problematic.

Rajan, S., et al. (2020). "Association of Symptoms of Depression With Cardiovascular Disease and Mortality in Low-, Middle-, and High-Income Countries." JAMA Psychiatry.

Also the risk of “bleeding” is very small—I personally have never come across it. By putting in bleeding the authors sound like a non-professional writers who is trying to hype things up. I find this really off-putting. I am less likely to take the article and its results seriously if the introduction has a “hype-up” flavour. I would take bleeding out and put in side-effects that are more common. In actual fact, bleeding is much more likely if a person takes aspirin or a non-steroidal medication than if they take an antidepressant.

As noted above I would also be very careful in making simplistic associations with cardiovascular disease.

Weighing up risks and benefits in medicine is always a difficult process and simplistic thinking does not facilitate it.

Line 38-119

This segment is written as one paragraph. Clearly this is too much information for one paragraph and it exhausting to read for the reader. I reads/feels like an “information dump”. The authors need to think about putting in breaks: breaking up this material into multiple smaller paragraphs that make the experience of reading more pleasant and manageable for the reader.

Line 110-111 RE: ARMT 

In a previous article psycho-regulatory massage therapy was given the acronym PRMT, here is it called psychoactive massage or psychoactive massage therapy,  or affect-regulatory massage therapy (ARMT). I think it is quite difficult for readers to manage multiple acronyms for the same thing. Is there consensus about acronyms or is this an area of disagreement between different Institutions in Germany?

Line 118

Here the authors use psychoactive massage therapy. I am not sure why the term affect-regulatory massage therapy (ARMT) was introduced before. I think the authors need to think about their terminology or they will mix the reader up.

Line 121

The authors now say “suitable massage treatment (ARMT)”, that is they oscillate between different terminologies. Also Suitable massage treatment ≠ ARMT

Line 184 RE: oral assessments

I think there is a translation problem here. But in English this sounds like they had their mouth examined. Maybe replace with Assessment on clinical interview?

Line 213 RE: Acute internal disease

I am unsure what this means? Does it mean a comorbid medical condition?

221 Re Within two months, 60 patients could be recruited.

I think this should read as “Within two months, 60 patients were recruited”.

239-240 RE: Techniques for psychoactive massage described in the literature, e.g. the Slow Stroke© Massage (28, 31) were taken as the matrix for development of our own massage technique described in detail below.

Both these references are in German. This means that English speaking massage therapists would be unable to replicate the technique. The article would be more useful if the there was information about what was actually done available via an appendix (in the article or online if this is possible) including a video demo which could be developed using one of the therapists as a patient. Currently the information is not usable for clinicians because the information is inaccessible unless the therapist can read German.

Line 300 RE: Disease-related data

As noted before depression is usually termed a disorder. Maybe the authors should just change the heading to something like Depression-related data.

Line 325 RE: somatic symptoms

I presume the authors mean functional somatic symptoms since all patients with a comorbid medical condition were excluded (or that is my understanding).  

Line 392 REL As outlined above, somatic symptoms such as psychomotor retardation, sleep disorders or 392 general fatigue are prominent features of the clinical picture of depression.

Here sleep disorders are added as a somatic symptom. As noted before the authors have to work out what they mean by somatic symptoms vs psychosomatic/functional somatic symptoms. The definition keeps changing and shifting.

Line 396 RE functional disorders

Here the authors use the term functional disorders whereas before they used psychosomatic. I think the manuscript needs to be checked for consistency. The authors need to decide on what terminology they want to use an to stick with it.

Line 312- 417

I wonder if the authors should think about putting some of this material in their introduction rather than here.

The first paragraph of the discussion should really summarise their findings so that the reader who does not want to trawl through the findings can get an quick overview of the study findings.

Again the paragraph here is far to long. No reader can maintain concentration for a long piece of text.

Second paragraph of the discussion

Again this paragraph is too long. Readers cannot manage this length.

Line 455 CT afferences

CT has never been defined I think. And afferences is a typo.

Line 463 RE: Practical application of this fundamental fact is reflected e.g. in the concept of "basal stimulation, opening up a path of communication in nursing e.g. in hospices that is otherwise blocked due to disability or serious illness

I think the authors are going off topic. The article is about depression.

Line 466 RE: Against the background of anhedonia described as the most common feature of depressive experience and behavior our findings are in some way “impossible”. How come that a depressed patients can feel and express bodily and verbally affective touch as a positive experience? Obviously empathic,  professional touch can slip into this basic communication disorder, i.e. can enter the disturbed emotional world of the patient. It fits into this concept that only participants of the massage group often expressed feelings of having been “accepted” in the final talks with the first author. The following English translation of a patient’s original final statement might help us in understanding some essentials of the therapeutic process:

I think the authors are musing and allowing themselves to express their inner thought processes. Clearly touch is important and I do not see this musing as helpful.

Line 474 This is qualitative results in the discussion section.

“I cried a lot the first time. For the first time I was able to feel my legs and my feet. Tears of joy 474 during the treatment. From the 2nd session onwards even better with a warming pad. Meeting of the 475 hands on my belly - very touching. I never experienced such a loving touch before. With the further 476 treatments I was loosening up completely. Already during the second session I perceived more than 477 the first time. Even my hands have been treated! Because of the depression I couldn't allow touching 478 me otherwise... My shell has become softer - my heart is opened. Massage therapist was totally 479 super gentle. Helped me to become a ´whole person´ again.”

Discussion

I had trouble getting through the discussion section. It needs to be rewritten so that the points are made in small clear paragraphs that the reader can digest.   There is much work to be done to tighten the discussion.

Author Response

Response to Reviewer (Reviewer 2)

We thank the reviewer for the comments and detailed proposals to amend this article. We accept many of the critical  statements. In the following we are going to describe the changes we have made and will also give our arguments where we were not able to follow the reviewer’s recommendations.

We would like to underline that the present version of the manuscript has been edited by a professional Editing Service of MDPI using British English. We can add the certificate of this service if needed.

In the following when describing what kind of changes have been made we shall always refer to the line numbers of this new edited version! When referring to special comments, we relate to the line numbers  of the original manuscript to which  the reviewer is referring to.

  • The term “somatic symptoms”: In fact we are referring to somatic symptoms that typically accompany depression. We deleted the term “psychosomatic” in the whole manuscript. (Except in line 122 where it is used in another context) We do not refer to “functional somatic symptoms” which acc. to the quoted paper by Roenneberg signify somatic symptoms in individuals without a clear-cut somatic or psychiatric diagnosis.
  • Abstract : anhedonia has not been mentioned.as an example of somatic symptoms but was added ( three typical symptoms separated by a comma.)
  • The abbreviation CT has been changed to C-tactile.
  • Line 39 (now line 40). We prefer the term disorder/ disease and have deleted ” illness”  in the whole manuscript.
  • Line 47: (now line 48) 10-15 % now comes after a semicolon instead of a full stop .
  • Line 49 (now line 50) brackets have been inserted: (e.g. with lithium salts)
  • Line 50 (now line 51): as explained above the term functional somatic symptoms would be inappropriate.
  • Line 53 (now 54): has been deleted.
  • Line 64 (now 69): has been changed accordingly.
  • Line 69 (now 73): has been changed.
  • Line 73: We do not think that we are quoting “simplistic associations“. The risks of antidepressants have become much clearer in recent years. This is a particularly sobering fact in view of thousands of patients receiving these drugs without appropriate indication. (as to the  impression of the reviewer that our critical sentences show a “hype –up flavor”  we may  annotate humbly  that the second author of this paper (BMOe) is an experienced psychopharmacologist and for 12 years  has been the chairman of the much respected  Drug Commission of the German Medical Association) The risk of bleeding is particularly increased in certain drug combinations. Nevertheless we have deleted “bleeding” in this context.
  • Line 38 -119 ; We have inserted various smaller paragraphs.
  • Line 110-111: Psychoactive massage is a term often used but not specified in a general, formal way. We use it as a general term comprising various concrete worked- out forms of massage therapy; all of them have the common goal to influence positively mental conditions. Examples are: Slow Stroke® Massage, Psychoregulatory Massage, Affect regulating Massage, Touch Life Massage etc.  Most of them take their origin from the Esalen massage. In a latter part of the paper it is explained that the Affect Regulating Massage is a special psychoactive technique developed by the first author. In Line 139 we have inserted the name of the special psychoactive massage used in the mentioned study (Slow Stroke massage).
  • Line 121 (now line 133): “suitable “ has been deleted.
  • Line 184 (now line 191): We follow the proposal of the reviewer.
  • Line 213 (now line 219): We have accepted the reviewer’s proposal.
  • Line 221 (now line 227): We have accepted the reviewer’s proposal
  • Line 240: The reviewer might have overlooked that the massage technique used in the present paper is described in detail (ref. new Lines 241 – 270.) Every skilled physiotherapist could  replicate the technique according to this description. The quoted German paper  by the way has an English abstract added.  We have also added hints to some video material from our collaborators)
  • Line 300 (now line 308): We have changed the heading to “Depression-related…. “
  • Line 329 (now line 333) : Somatic symptoms: This is the original term in the HAMD scale. Cannot be changed.  We do not mean functional somatic symptoms.
  • Line 392 (now line 396) : For our understanding our terminology is sufficiently clear and unambiguous . e.g. sleep disorder or fatigue are well known somatic symptoms to be found in many depressed patients.
  • Line 396 (now line 402):here we refer to the argumentation of                      Thomas Fuchs. No change.
  • Line 312 (the reviewer obviously is referring to line 396 to 417.)

It was by purpose that we did not want to make the introduction too                  heavy and long in order to keep it digestible. We have therefore clearly split         the arguments and facts presented in introduction and discussion.

We would also like to make the reviewer aware of  the last part of the paper, the conclusion, where the main results are presented in a compact form.

The reviewer is critical about the overall length of the discussion, which by the    way would be even longer in case we would start it with a summary of the       results. We have therefore inserted some more breaks between the various         aspects of mechanisms involved. We have also seriously checked whether   complete parts of the discussion could be removed. However we had to            conclude that this would seriously destroy the inner structure of our             argumentation. But having it split up in now  11 sections we feel that the text       will be better digestible.

  • Line 463: we agree with the critic and have deleted this sentence incl. the reference .
  • Line 466 (now line 479): The term “impossible” has been replaced by “paradoxical” which should make the meaning clearer.
  • The “musing” of the authors should help readers to understand what is actually happening in the treated individuals on the level of experience and behavior , apart from neurobiological reasoning which often has a  speculative character and which is presented  in many other papers.  We are not aware of authors except those few quoted here who actually gave some effort to this rather phenomenological approach.
  • Line 474: Qualitative data: The title of the paper “controlled mixed-methods study” refers to the fact that also qualitative data have been taken into consideration, a very special feature of this paper.

Thanks again for  the careful reviewing of our paper,- in which we have put special effort  in order to make the antidepressive effects of therapeutic touch better understood , in our common effort to improve the treatment of our depressed patients.

Round 2

Reviewer 1 Report

The authors have addressed all points (raised by this reviewer) satisfactorily.